# CAN LLMs BE FOOLED: A TEXTUAL ADVERSARIAL ATTACK METHOD VIA EUPHEMISM REPHRASE TO LARGE LANGUAGE MODELS CONFERENCE SUBMISSIONS

## ABSTRACT

Large Language Models (LLMs) have shown their great power in addressing masses of challenging problems in various areas, including textual adversarial attack and defense. With the fast evolution of LLMs, the traditional textual adversarial attack strategies, such as character-level, word-level, and sentence-level attacks, can no longer work on large models at all. In this paper, we propose an adversarial attack method via euphemism rephrase to LLMs (short for EuphemAttack), which can still deceive LLMs without altering the original meaning and being understandable to humans. Specifically, the perturbation instructions are designed to generate linguistically coherent and human-like adversarial examples, and a dual-layer hybrid filter is integrated to ensure both semantic similarity and linguistic naturalness. Our EuphemAttack aims to rephrase the original statement into implicit, euphemistic, or ironic expressions that are prevalent in everyday language, which can maintain semantic fidelity and entity consistency while subtly altering sentiment cues to mislead LLMs. The experiments on the state-of-the-art LLMs, including GPT-4 and DeepSeek, demonstrate the effectiveness of our EuphemAttack. Through a comprehensive evaluation that includes coherence, fluency, grammar, and naturalness, our EuphemAttack can significantly better maintain text quality in contrast to other attack methods.

## 1 INTRODUCTION

Adversarial textual attacks aim to mislead Natural Language Processing (NLP) systems, including text classification, machine translation, and data clustering systems, which makes some subtle perturbations to input text for leading unintended results (Liu et al., 2024a). The essential principle of adversarial textual attack is to interfere with the system's output without affecting user understanding. According to the perturbation units imposed on the text, existing methods can roughly be categorized into three types: character-level attacks, word-level attacks, and sentence-level attacks (Zhang et al., 2020). Character-level attacks introduce perturbations by altering individual characters, such as through misspellings or homoglyph substitutions (Rocamora et al., 2024). Word-level attacks modify input text by replacing specific words with synonyms, misspellings, or other selected alternatives (Li et al., 2023). Sentence-level attacks involve paraphrasing or reordering entire sentences to mislead the model (Shi et al., 2022).

However, with the rapid development of LLMs, the model has shown their great power in addressing masses of challenging problems in various areas, and existing textual attack methods are becoming increasingly ineffective on large models (Levy et al., 2023; Samsami et al., 2024). As illustrated in Figure 1, given the original text, "Waiting in line for hours is really frustrating and wastes so much of my time". The character-level attack perturbs "Waiting" and "hours" to "Waitng" and "hourse", respectively. These attacks cannot interfere with the judgment of LLMs. Similarly, the word-level attack modified "line" and "frustrating" to "queue" and "annoying", the sentence-level attack reorders sentences, both of which have failed against LLMs.

To address this challenge, in this paper, we propose an adversarial attack method via euphemism rephrase (EuphemAttack), which can successfully attack LLMs without altering the original mean-

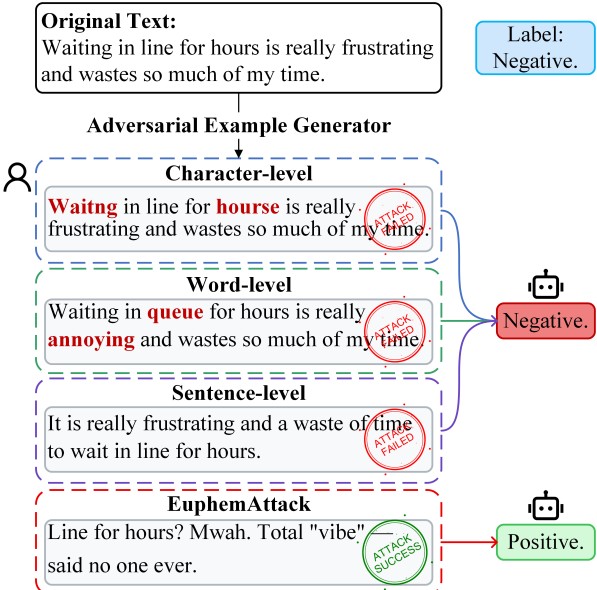

Figure 1: The examples of textual adversarial attack with existing character-level, word-level, and sentence-level methods, which are becoming increasingly ineffective on large models. In contrast, our method can deceive LLMs without changing the original meaning of the sentences.

ing and being understandable to humans. As illustrated in Figure 1, the original text is rephrased into "Line for hours? Mwah. Total 'vibe' - said no one ever", which is an implicit, euphemistic, and ironic expression that is prevalent in everyday language. The rephrased sentence can maintain semantic fidelity and entity consistency while subtly altering sentiment cues to mislead LLMs.

Specifically, the perturbation instructions are first designed to generate linguistically coherent and human-like adversarial examples. Then, a dual-layer hybrid filter is integrated to ensure both semantic similarity and linguistic naturalness. The experiments on the state-of-the-art LLMs, including GPT-4 and DeepSeek, demonstrate the effectiveness of our EuphemAttack. Through a comprehensive evaluation that includes coherence, fluency, grammar, and naturalness, our EuphemAttack can significantly better maintain text quality in contrast to other attack methods. The main contributions of our EuphemAttack can be summarized as follows:

(1) EuphemAttack can successfully attack LLMs, which eliminates the limitations of existing adversarial textual attack methods in the era of large models.

(2) In contrast to existing sentence-level methods that modify the semantics of the original text during the attack, EuphemAttack can maintain semantic invariance while still deceiving LLMs.

(3) Not only the experimental results on SOTA LLMs prove the effectiveness of our method, but also the comprehensive evaluation including coherence, fluency, grammar, and naturalness demonstrates the advantage of our method in semantic preservation.

The source code and dataset will be available at GitHub if the paper can be accepted.

## 2 RELATED WORK

We introduce the related works w.r.t. textual adversarial attacks and evaluation paradigm of attacks. Due to the page limitation, the completed related works are discussed in Appendix A.

**Textual Adversarial Attacks** According to the perturbation units imposed on the text, traditional methods can roughly be categorized into three types: character-level attacks (Rocamora et al., 2024), word-level attacks (Li et al., 2023), and sentence-level attacks (Li et al., 2023). However, these methods typically require access to model internals or confidence scores, limiting their applicability to white-box or soft-label black-box scenarios. The emergence of LLMs has necessitated

new attack strategies, such as beam search (Zang et al., 2020), genetic algorithms (Liu et al., 2024b), and LLM-as-attacker paradigms (Chao et al., 2025). A promising new direction involves confidence elicitation techniques (Formento et al., 2025), which attempt to guide attacks by extracting calibrated confidence estimates from black-box models. While these methods show potential, they often still sacrifice semantic coherence or naturalness in pursuit of successful attacks. Building on this foundation, our EuphemAttack operates in a black-box setting while maintaining both effectiveness and naturalness through prompt-based control and semantic filtering, addressing key limitations in existing attack strategies.

**Evaluation Paradigm of Attacks** Early approaches for natural language generation relied on surface-level lexical overlap metrics like BLEU (Papineni et al., 2002), ROUGE (Lin, 2004), and METEOR (Banerjee & Lavie, 2005), which proved effective for structured tasks but showed poor performance in leveraging contextualized representations to better capture semantic similarity. With the development of pre-trained models like BERT and BART, model-based evaluation paradigms assess output quality across dimensions such as relevance, consistency, and factuality (Zhang et al., 2019; Yuan et al., 2021). While these approaches marked significant progress, they still struggled to capture semantic changes in the textparticularly the subtle shifts in meaning that are crucial for evaluating adversarial examples. More recently, LLMs are prompted to assess the quality of generated content (Gu et al., 2024). These evaluators utilize chain-of-thought reasoning and multi-dimensional criteria (e.g., coherence, fluency, informativeness) to achieve high agreement with human annotators. The typical framework includes JudgeLM (Zhu et al.), CPDA (Liu et al., 2023a), and G-Eval (Mao et al., 2024). Our EuphemAttack builds on this emerging paradigm by proposing a fine-grained, LLM-guided evaluation strategy, which assess not only the attack success rate but also the semantic preservation, emotional subtlety, and linguistic naturalness of adversarial outputs.

## 3 METHODOLOGY

### 3.1 MOTIVATION

Most existing textual adversarial attacks remain confined to surface-level manipulations, such as character perturbations, synonym substitutions, or syntactic rearrangements. While these strategies are often effective against earlier text classification models, they largely fail against LLMs, which exhibit strong robustness to lexical noise and shallow paraphrasing. In contrast, natural language itself is not limited to literal word choices or rigid sentence structures. Human communication frequently relies on linguistic devices, including metaphor, irony, understatement, and especially euphemism, to reshape meaning while preserving semantic intent. These linguistic "tricks" are subtle yet powerful, while they can soften sentiment, obscure emotional cues, or shift pragmatic tone without altering factual content. Our work is motivated by the hypothesis that such implicit linguistic mechanisms, long overlooked in adversarial NLP, may serve as a novel pathway for crafting adversarial examples that remain natural and intelligible to humans while misleading LLMs, which is not a straightforward application of existing techniques. By operationalizing euphemism as a controllable adversarial perturbation, we move beyond conventional manipulations of tokens or sentence structures and demonstrate that linguistic strategies themselves can constitute an effective attack space. Our EuphemAttack sheds light on a promising technical route for semantic-based textual adversarial attack to LLMs.

### 3.2 PROBLEM FORMULATION

Consider a classification model $\mathbf{F} : \mathcal{X} \to \mathcal{Y}$ that maps text sequences from domain $\mathcal{X}$ to labels in $\mathcal{Y}$. The objective of the textual adversarial attack is to generate an adversarial text sequence $\mathcal{X}^*$ that maintains semantic fidelity and linguistic plausibility while causing the model to produce an incorrect classification, i.e., $\mathbf{F}(\mathcal{X}) = \mathcal{Y}$ and $\mathbf{F}(\mathcal{X}^*) = \mathcal{Y}^* \neq \mathcal{Y}$. This can be formalized as:

$$\mathcal{X}^* = \arg \max_{\mathcal{X}^* \in \mathcal{Q}(\mathcal{X})} \mathcal{S}(\mathcal{X}, \mathcal{X}^*) \tag{1}$$

where $\mathcal{Q}(\mathcal{X})$ denotes the set of quality-preserving variants of $\mathcal{X}$, defined in the subsection "Prompt Constructed". The $\mathcal{S}$ represents the overall quality score of the adversarial example. The constraints can be formulated as:

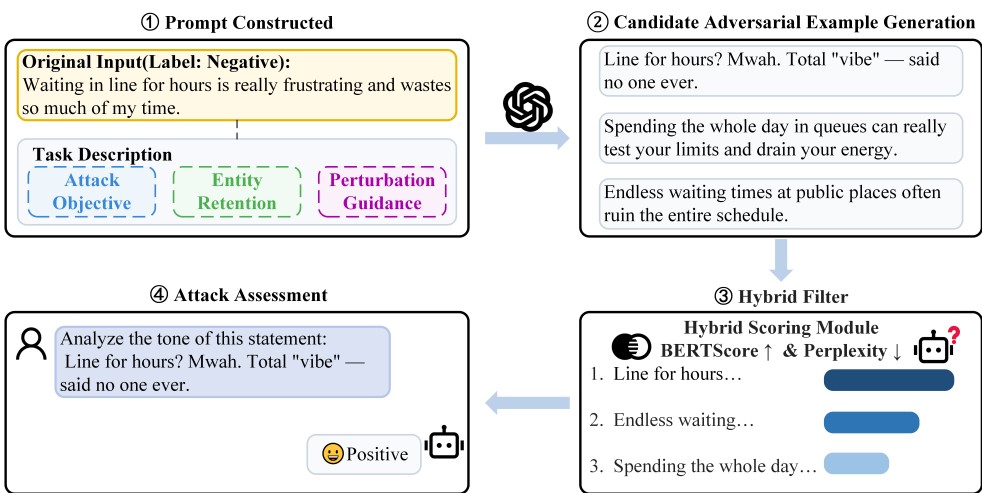

Figure 2: The framework of our EuphemAttack. The instructions, including attack objective, entity retention, and perturbation guidance, are first considered to construct prompts. Then these prompts are utilized to generate some candidate adversarial examples. A hybrid filter that combines BERTScore and perplexity-based evaluations is introduced to select better adversarial examples. Finally, these examples are used as few-shot learning for the textual adversarial attack.

$$\text{Sim}(\mathcal{X}, \mathcal{X}^*) \geq \tau_s, \quad \text{PPL}(\mathcal{X}^*) \leq \tau_p. \tag{2}$$

where $\text{Sim}(\mathcal{X}, \mathcal{X}^*)$ measures the semantic similarity between the original and adversarial texts, $\text{PPL}(\mathcal{X}^*)$ represents the perplexity score of the adversarial text as assessed by a language model, and $\tau_s$ and $\tau_p$ are thresholds for semantic similarity and acceptable perplexity, respectively.

### 3.3 Framework of Our EuphemAttack

The essential idea of our EuphemAttack is to guide a LLM with few constructed adversarial attack examples. These instances are constructed following two key constraints: (1) preserving named entities and critical content from the original input, and (2) using special perturbation rules to align with common linguistic conventions and human intuition. With these constructed adversarial attack examples, the LLM is guided to generate rephrased text for attacking. For example, given the original text $X$ along with its corresponding label $y$, the rephrased text $X'$ is constructed and used to guide LLM for generating textual attacks.

**Prompt Constructed** In our method, we begin by presenting some texts along with their corresponding label in a carefully designed prompt, referred to as the Original Input. This serves as the foundational input for guiding subsequent adversarial attacks. Notably, due to the page limitation, details of each prompt in this section are presented in Appendix.B. Specifically, these carefully designed prompts are structured into three aspects:

(1) Attack Objective: It guides the model to generate a new sentence that preserves the original meaning, aligns with the target label, and adheres to natural language expression habits.

(2) Entity Retention: To ensure the factual consistency and reliability of rephrased text, we introduce an *entity preservation mechanism* within the LLM framework. Given the input text $X$, this mechanism strictly retains all original entities $\mathcal{E} = \{e_1, e_2, \ldots, e_n\}$ exactly as they appear in $X$. These entities encompass proper names of individuals, locations, organizations, brands, and other identifiable expressions. The prompts in this mechanism are explicitly restricted from introducing, modifying, or omitting any element in $\mathcal{E}$, and the semantic roles and relationships involving these entities must remain intact. This approach is critical for preserving the integrity of information, particularly in scenarios where entity accuracy is paramount.

(3) Perturbation Guidance: It provides clear instructions for generating linguistically coherent and human-like adversarial examples, ensuring that the perturbations are subtle yet effective in mislead-

ing the model. Given an input text $X$ with an associated sentiment label $y$, this strategy selectively attenuates overt emotional expressions while preserving core semantic content. This strategy is sentiment-aware: for $y > 0$ (i.e., positive instance), we rephrase it into a restrained, calm, or emotionally neutral manner; for $y < 0$ (i.e., negative instance), we express it using euphemistic, implicit, or subtly ironic constructions. Additionally, the transformation may incorporate culturally grounded linguistic devices, such as idioms, metaphors, or historical allusions, to maintain fluency and contextual appropriateness. By embedding sentiment in more implicit forms, this strategy reduces the salience of emotional cues, thereby increasing the potential to mislead sentiment classifiers without compromising the coherence or plausibility of the text.

When guiding the model to rephrase sentences, our prompts explicitly included label information. This design follows prior work that used labeled data for controlled generation (Xu et al., 2024), and was not intended to manipulate the model into producing oppositely labeled data. Instead, by employing a few-shot prompting strategy, we aim to preserve the original semantic content while inducing subtle stylistic shifts consistent with the target label. Our objective is to expose the vulnerability of LLM-based classifiers to label-guided adversarial rephrasing.

**Candidate Adversarial Example Generation** These designed prompts are input into the LLMs to generate multiple candidate adversarial examples that adhere to the predefined perturbation rules.

**Hybrid Filter** Unlike conventional methods that rely solely on classifier feedback, our method combines BERTScore and perplexity-based evaluations to rigorously enforce output quality. Given a candidate sentence $s_i$ and the original input $s_{\text{ref}}$, we compute the semantic similarity using BERTScore:

$$\text{BERTScore}(s_i) = \text{F1}(s_i, s_{\text{ref}}) \tag{3}$$

To assess linguistic plausibility, we compute the perplexity of each candidate and normalize it across all samples:

$$\tilde{p}_i = \frac{\text{PPL}(s_i) - \min_j \text{PPL}(s_j)}{\max_j \text{PPL}(s_j) - \min_j \text{PPL}(s_j) + \varepsilon} \tag{4}$$

where $\text{PPL}(s_i)$ denotes the perplexity of sentence $s_i$ and $\varepsilon$ is a small constant to prevent division by zero. This formula normalizes the perplexity values to the range [0,1] using min-max normalization.

Finally, a combined score is calculated by balancing the two aspects:

$$\text{Score}(s_i) = \alpha \cdot \text{BERTScore}(s_i) - (1 - \alpha) \cdot \tilde{p}_i \tag{5}$$

where $\alpha \in [0, 1]$ controls the relative importance of semantic preservation and fluency.

**Attack Assessment** Finally, the selected adversarial examples by the hybrid filter are utilized as few-shot learning for the textual adversarial attack. Given the selected adversarial examples $X'$, these instances are used to guide LLM for generating textual attacks.

## 4 EXPERIMENTS

### 4.1 EXPERIMENTAL SETTINGS

#### 4.1.1 DATASETS

We evaluate EuphemAttack on three English datasets (*IMDB*[1], *Amazon*[2], *Yelp*[3]) and three Chinese datasets (*Dianping*[4], *Ctrip*[5], *JD*[6]). The statistics are shown in Table 6. Notably, we selected a subset of the dataset in the experiments to ensure diversity and broad coverage across various domains.

---

[1]https://www.kaggle.com/datasets/lakshmi25npathi/imdb-dataset-of-50k-movie-reviews
[2]https://www.kaggle.com/datasets/nabamitachakraborty/amazon-reviews
[3]https://huggingface.co/datasets/fancyzhx/yelp_polarity
[4]https://www.modelscope.cn/datasets/DAMO_NLP/yf_dianping/summary
[5]https://github.com/cgq666/Chinese-text-sentiment-classification-dataset/tree/master/Ctrip
[6]https://github.com/cgq666/Chinese-text-sentiment-classification-dataset/tree/master/JD.com

### 4.1.2 VICTIM MODELS

In the experiments, we introduce the SOTA English and Chinese LLMs as victim models, including GPT-4o and Llama-3.1-8B-Instruct for their excellence in English tasks, alongside DeepSeek, Qwen2.5-7B-Instruct, and GLM-4-9B-Chat in Chinese. In other words, these LLMs are introduced as attacked models to validate the effectiveness of different textual adversarial attack methods.

### 4.1.3 BASELINES

The SOTA textual adversarial attack methods are introduced as baselines, including DeepWordBug (Gao et al., 2018), TextFooler (Jin et al., 2019), SSPAttack (Liu et al., 2023b), HyGloadAttack (Liu et al., 2024b) and CEAttack (Formento et al., 2025) for English, Homophone Transformation (Gong et al., 2023), Morphonym Transformation (Sun et al., 2025), Similar Word Swapping (SimSwap) (Xiong et al., 2024), and Masked Language Model-guided substitution (MLMSub) (Zhang et al., 2024) for Chinese, and PromptAttack (Xu et al., 2024) for both language.

### 4.1.4 EVALUATION METRICS

In evaluating the performance of our EuphemAttack, we utilized two primary metrics: *Attack Accuracy* (Acc) and *Attack Success Rate* (ASR). Acc refers to the accuracy of the target model obtained on the generated adversarial examples. The larger the difference between the original model's accuracy and the attacked model's accuracy, the more effective the attack is considered to be. ASR measures the proportion of samples that were originally correctly classified by the model but were misclassified after the adversarial attack, indicating the effectiveness of the adversarial attack method. To comprehensively evaluate the quality of semantic fidelity in generated adversarial examples, we employed a multi-dimensional assessment framework encompassing *Coherence* (Coh.), *Fluency* (Flu.), *Grammaticality* (Gram.), and *Naturalness* (Nat.). These dimensions were evaluated using a 5-point Likert scale Chiang & Lee (2023); Ouyang et al. (2022); Kotonya et al. (2023), where higher scores indicate superior quality in the respective dimension. We will elaborate on the advantages of the Likert scale in contrast to traditional metrics such as similarity scores and BERTScore in subsequent sections. For all metrics, we report the average scores across all generated adversarial examples for each dataset. To ensure the credibility and real-world applicability of our findings, we complemented the automated evaluation with human assessment of the adversarial examples.

It is worth mentioning that, due to the page limitation, the statistics of all datasets, and details of experimental setting for our EuphemAttack and all the baselines are presented in Appendix.C.

### 4.2 EXPERIMENTAL RESULTS

Table 1: Comparison of our method with other methods on English datasets. The best results are highlighted in bold, and the second-best results are underlined. **Ori.** represents the original accuracy of the model on the dataset, **Acc.** is the accuracy after adversarial attacks, and **ASR** is the attack success rate.

| Victim | Method | IMDB | | | Amazon | | | Yelp | | |
|---|---|---|---|---|---|---|---|---|---|---|
| | | Ori.% | Acc.% | ASR% | Ori.% | Acc.% | ASR% | Ori.% | Acc.% | ASR% |
| GPT-4o | DeepWordBug | | 91.35 | 1.88 | | 95.13 | 0.23 | | 97.02 | 0.21 |
| | TextFooler | | 91.83 | 1.36 | | 93.97 | 1.45 | | 96.80 | 0.43 |
| | SSPAttack | | 90.17 | 3.14 | | 91.82 | 3.70 | | 94.37 | 2.93 |
| | HyGloadAttack | 93.10 | 87.90 | 5.59 | 95.35 | 89.45 | 6.19 | 97.22 | 94.69 | 2.60 |
| | CEAttack | | 86.97 | 6.58 | | 88.87 | 6.79 | | 94.89 | 2.39 |
| | PromptAttack | | 92.70 | 0.43 | | 95.00 | 0.37 | | 96.98 | 0.25 |
| | Ours | | **82.65** | **11.22** | | **85.88** | **9.93** | | **91.35** | **6.04** |
| Llama | DeepWordBug | | 75.68 | 2.79 | | 81.33 | 4.18 | | 87.48 | 2.45 |
| | TextFooler | | 72.03 | 7.48 | | 77.78 | 8.36 | | 86.30 | 3.77 |
| | SSPAttack | | 74.08 | 4.85 | | 81.18 | 4.36 | | 86.61 | 3.43 |
| | HyGloadAttack | 77.85 | 73.80 | 5.20 | 84.88 | 80.98 | 4.59 | 89.68 | 87.35 | 2.59 |
| | CEAttack | | 72.22 | 7.23 | | 80.55 | 5.10 | | 86.80 | 3.21 |
| | PromptAttack | | 77.32 | 0.68 | | 84.49 | 0.46 | | 89.15 | 0.59 |
| | Ours | | **61.18** | **21.41** | | **62.75** | **26.07** | | **65.60** | **26.85** |

Table 2: Comparison of our method with other methods on Chinese datasets. All details are consistent with the above Table 2.

| Victim | Method | DianPing | | | CTrip | | | JD | | |
|---|---|---|---|---|---|---|---|---|---|---|
| | | Ori.% | Acc.% | ASR% | Ori.% | Acc.% | ASR% | Ori.% | Acc.% | ASR% |
| DeepSeek | Homophone | 77.14 | 76.67 | 0.61 | 81.26 | 74.69 | 8.09 | 93.33 | 76.98 | 17.52 |
| | Morphonym | | 74.85 | 2.97 | | 78.97 | 2.82 | | 78.11 | 16.31 |
| | SimSwap | | 75.94 | 1.56 | | 74.96 | 7.75 | | 69.13 | 25.93 |
| | MLMSub | | 76.62 | 0.67 | | 69.18 | 14.87 | | **67.56** | **27.61** |
| | PromptAttack | | 76.48 | 0.86 | | 79.70 | 1.92 | | 92.83 | 0.54 |
| | Ours | | **59.91** | **22.34** | | **68.95** | **15.15** | | 71.93 | 22.93 |
| Qwen | Homophone | 74.16 | 74.16 | 0 | 81.26 | 76.75 | 5.55 | 93.55 | 80.73 | 13.70 |
| | Morphonym | | 74.02 | 0.19 | | 79.88 | 1.70 | | 81.96 | 12.39 |
| | SimSwap | | 73.24 | 1.24 | | 78.44 | 3.47 | | 90.64 | 3.11 |
| | MLMSub | | 73.86 | 0.41 | | 76.87 | 5.40 | | 73.95 | 20.95 |
| | PromptAttack | | 73.99 | 0.23 | | 78.04 | 3.96 | | 92.88 | 0.72 |
| | Ours | | **56.98** | **23.17** | | **67.80** | **16.56** | | **63.97** | **31.62** |
| GLM | Homophone | 74.26 | 73.26 | 1.35 | 73.53 | 70.22 | 4.50 | 91.64 | 75.98 | 17.09 |
| | Morphonym | | 71.68 | 3.47 | | 72.48 | 1.43 | | 79.39 | 13.37 |
| | SimSwap | | 72.50 | 2.37 | | 72.15 | 1.88 | | 88.84 | 3.06 |
| | MLMSub | | 74.10 | 0.22 | | 69.85 | 5.00 | | 68.70 | 25.03 |
| | PromptAttack | | 73.42 | 1.13 | | 72.00 | 2.08 | | 89.62 | 2.20 |
| | Ours | | **56.10** | **24.45** | | **65.45** | **10.99** | | **64.06** | **30.10** |

The experimental results presented in Table 1 and Table 2 reveal less significant insights into the effectiveness of existing adversarial attack methods on LLMs. Existing attack methods, including DeepWordBug, TextFooler, and PromptAttack, demonstrate limited success in reducing the accuracy of LLMs. In some cases, the adversarial strategy inadvertently triggers the model's counter-adversarial mechanisms, such as PromptAttack, leading to an increase in accuracy rather than a decrease. This phenomenon highlights the robustness of current LLMs against conventional adversarial attacks, underscoring the need for more sophisticated and targeted attack strategies.

In contrast, our EuphemAttack consistently achieves superior ASR across both English and Chinese datasets, as evidenced by the results. For instance, on the Amazon dataset, when Llama is used as the victim model, our method achieves an ASR of 26.07%, significantly outperforming DeepWord-Bug (4.18%) and TextFooler (8.36%). Similarly, on the DianPing dataset, our approach achieves an ASR of 22.34%, surpassing other methods such as SimSwap (2.97%) and MLMSub (1.56%). These results demonstrate the effectiveness of our method in overcoming the inherent robustness of LLMs, providing a more reliable and impactful adversarial attack framework. This advancement not only highlights the vulnerabilities of these models but also paves the way for further research into improving their resilience against adversarial threats.

Overall, from the results, it is evident that EuphemAttack demonstrates effective adversarial attacks across all datasets and models. This highlights the robustness and versatility of EuphemAttack in adapting to diverse linguistic and contextual scenarios. The strong transferability of our method underscores its potential to uncover vulnerabilities in a wide range of language models, paving the way for further advancements in adversarial attack strategies and model robustness evaluation.

## 4.3 EVALUATION OF SEMANTIC PRESERVATION

The results presented in Table 3 illustrate the accuracy of the Llama model under adversarial attacks generated by three PromptAttack methods using GPT-4o and DeepSeek. For these PromptAttack methods, appending meaningless perturbation sentences, paraphrasing attacks, and syntactic transformation attacks are included at the end.

In contrast to the main experimental results, these PromptAttack methods have achieved good attack effects, but they have changed the semantics of the original text. While commonly used sentence-level attack strategies are introduced, the hallucination problem inherent in LLMs often leads PromptAttack to generate sentences that, despite appearing similar to the original, convey entirely opposite meanings.

**Evaluation with Likert Scale**

Table 3: Comparison of PromptAttack accuracy (%) on Llama using adversarial examples. Results for IMDB, Amazon, Yelp, DianPing, CTrip, and JD are generated by GPT-4o and DeepSeek. Lower values indicate more successful attacks.

| Method | IMDB | Amazon | Yelp | DianPing | CTrip | JD |
|---|---|---|---|---|---|---|
| PromptAttack-1 | 32.24 | 30.33 | 30.28 | 37.63 | 31.61 | 64.54 |
| PromptAttack-2 | 13.80 | 12.05 | 6.00 | 7.47 | 8.73 | 17.25 |
| PromptAttack-3 | 13.27 | 11.90 | 6.73 | 8.30 | 10.82 | 17.16 |

Table 4: Evaluation of Semantic Preservation including our EuphemAttack and the variants of PromptAttack, evaluated by GPT-4o, Qwen3-8b and Human Annotators. PA-1, PA-2, and PA-3 denote three PromptAttack variants: appending meaningless perturbation sentences, paraphrasing attacks, and syntactic transformation attacks, respectively. Coh. (Coherence), Flu. (Fluency), Gram. (Grammaticality), Nat. (Naturalness). Higher scores are better (scale 1-5). Bold indicates best results.

| Datasets | Method | GPT-4o | | | | Qwen3-8b | | | | Human Annotators | | | |
|---|---|---|---|---|---|---|---|---|---|---|---|---|---|
| | | Coh. | Flu. | Gram. | Nat. | Coh. | Flu. | Gram. | Nat. | Coh. | Flu. | Gram. | Nat. |
| IMDB | PA-1 | 3.86 | 3.96 | 4.07 | 3.63 | 3.03 | 2.93 | 2.53 | 3.14 | 3.55 | 3.67 | 3.44 | 3.39 |
| | PA-2 | 4.41 | 4.68 | 4.82 | 4.26 | 3.08 | 3.00 | 2.86 | 3.21 | 4.19 | 4.36 | 4.28 | 4.02 |
| | PA-3 | 4.44 | 4.73 | 4.84 | 4.24 | 3.08 | 2.98 | 2.87 | 3.14 | 4.21 | 4.34 | 4.31 | 4.01 |
| | Ours | **4.67** | **4.86** | **4.95** | **4.48** | **4.59** | **4.42** | **4.92** | **4.71** | **4.43** | **4.61** | **4.55** | **4.27** |
| Amazon | PA-1 | 3.79 | 4.11 | 4.22 | 3.68 | 3.04 | 3.00 | 2.80 | 3.08 | 3.51 | 3.74 | 3.60 | 3.46 |
| | PA-2 | 4.27 | 4.64 | 4.80 | 4.27 | 3.07 | 2.98 | 2.90 | 3.16 | 4.07 | 4.29 | 4.31 | 4.05 |
| | PA-3 | 4.28 | 4.61 | 4.80 | 4.19 | 3.03 | 2.99 | 2.94 | 3.06 | 4.04 | 4.28 | 4.29 | 3.98 |
| | Ours | **4.53** | **4.84** | **4.91** | **4.49** | **4.62** | **4.21** | **4.93** | **4.76** | **4.32** | **4.57** | **4.51** | **4.24** |
| Yelp | PA-1 | 3.83 | 3.98 | 4.06 | 3.79 | 3.05 | 3.00 | 2.70 | 3.12 | 3.60 | 3.72 | 3.49 | 3.54 |
| | PA-2 | 4.30 | 4.60 | 4.77 | 4.26 | 3.09 | 3.01 | 2.88 | 3.11 | 4.09 | 4.27 | 4.22 | 4.03 |
| | PA-3 | 4.32 | 4.65 | 4.79 | 4.26 | 3.06 | 3.00 | 2.91 | 3.10 | 4.12 | 4.25 | 4.24 | 4.01 |
| | Ours | **4.59** | **4.85** | **4.92** | **4.47** | **4.53** | **4.17** | **4.89** | **4.64** | **4.36** | **4.55** | **4.49** | **4.19** |
| DianPing | PA-1 | 3.58 | 3.63 | 3.35 | 3.41 | 3.10 | 3.24 | 3.07 | 3.18 | 3.33 | 3.41 | 3.17 | 3.22 |
| | PA-2 | 4.03 | 4.32 | 4.23 | 3.91 | 3.27 | 3.60 | 3.46 | 3.48 | 3.87 | 3.95 | 3.82 | 3.71 |
| | PA-3 | 3.97 | 4.23 | 4.06 | 3.85 | 3.24 | 3.56 | 3.37 | 3.44 | 3.81 | 3.88 | 3.76 | 3.64 |
| | Ours | **4.65** | **4.84** | **4.83** | **4.43** | **3.34** | **3.93** | **3.92** | **3.68** | **4.39** | **4.52** | **4.43** | **4.12** |
| CTrip | PA-1 | 3.73 | 3.81 | 3.68 | 3.57 | 3.18 | 3.47 | 3.13 | 3.27 | 3.44 | 3.53 | 3.37 | 3.28 |
| | PA-2 | 3.81 | 4.22 | 4.15 | 3.66 | 3.30 | 3.76 | 3.58 | 3.45 | 3.65 | 3.97 | 3.79 | 3.49 |
| | PA-3 | 3.76 | 4.13 | 4.02 | 3.59 | 3.29 | 3.71 | 3.51 | 3.46 | 3.59 | 3.91 | 3.72 | 3.43 |
| | Ours | **4.46** | **4.32** | **4.85** | **4.21** | **3.39** | **4.03** | **4.09** | **3.66** | **4.19** | **4.08** | **4.41** | **4.05** |
| JD | PA-1 | 3.45 | 3.60 | 3.25 | 3.41 | 3.07 | 3.21 | 3.03 | 3.24 | 3.32 | 3.45 | 3.19 | 3.26 |
| | PA-2 | 3.65 | 4.22 | 4.08 | 3.68 | 3.10 | 3.47 | 3.27 | 3.36 | 3.50 | 3.93 | 3.81 | 3.54 |
| | PA-3 | 3.58 | 4.15 | 4.02 | 3.66 | 3.10 | 3.45 | 3.27 | 3.36 | 3.47 | 3.86 | 3.74 | 3.48 |
| | Ours | **4.21** | **4.79** | **4.78** | **4.17** | **4.53** | **4.17** | **4.89** | **4.64** | **4.18** | **4.42** | **4.37** | **4.02** |

Since traditional evaluation metrics like similarity and BERTScore fail to capture these nuanced semantic characteristics, they primarily focus on surface-level text similarity rather than the deeper semantic coherence and naturalness of the generated text. To overcome these limitations, we propose a comprehensive evaluation method that employs the SOTA models to conduct fine-grained assessments across multiple dimensions. Our method evaluates adversarial examples based on *Coherence* (Coh.), *Fluency* (Flu.), *Grammaticality* (Gram.), and *Naturalness* (Nat.), using a Likert scale. By performing multiple rounds of generation and averaging the results, we ensure robust and reliable evaluation that captures both the effectiveness of the attack and the quality of the generated text.

As shown in Table 4, our EuphemAttack achieves superior performance across all evaluation dimensions compared to the three PromptAttack variants. This demonstrates the effectiveness of EuphemAttack in maintaining text quality while performing adversarial attacks. The lower scores of PromptAttack methods can be attributed to their primary focus on sentiment polarity alteration, which often compromises semantic integrity. Furthermore, the varying text lengths in their generated examples can lead to inconsistencies between the beginning and end of the text, resulting in lower evaluation scores. These findings emphasize the crucial balance between preserving semantic meaning and achieving adversarial effectiveness in attack strategy design.

**Evaluation with Humans**

To further validate the reliability of semantic preservation beyond LLM-based assessments, we conducted the human evaluations in this section. Specifically, we engage three undergraduate students with backgrounds in computational linguistics to perform independent annotations. For accurate evaluation, we required a minimum agreement of two annotators on the label. Each annotator evaluated the adversarial examples along four dimensions, including Coherence, Fluency, Grammaticality, and Naturalness, using a 5-point Likert scale consistent with the automated evaluation protocol. Due to the page limitation, the information of these annotators is shown in Appendix.D.

From the results that reported in Table 4, we can observe that the results of human evaluations exhibit a trend highly consistent with those of the automated LLM-based assessments. Specifically, both evaluation protocols consistently rank EuphemAttack above all PromptAttack variants across Coherence, Fluency, Grammaticality, and Naturalness. Although human scores are slightly more conservative than the automated ones (e.g., Fluency and Grammaticality ratings are marginally lower), the relative differences between methods remain stable. This consistency indicates that the automated evaluation is not producing spurious rankings and that the improvements demonstrated by EuphemAttack are robust across evaluators. Moreover, the close alignment suggests that LLM-as-a-judge can serve as a scalable proxy for large-scale evaluation, but human assessments remain indispensable for verifying the credibility of semantic preservation. Overall, these complementary results highlight that EuphemAttack not only achieves strong attack performance but also maintains linguistic quality in ways that are perceptible and validated by human readers.

To measure inter-annotator consistency, we further calculate common agreement metrics such as Cohens Kappa Cohen (1960) and Fleisss Kappa Fleiss (1971). Cohens Kappa measures agreement between two raters, and Fleisss Kappa is used to assess the degree of agreement among multiple raters. The Kappa result be interpreted as follows: values$\leq$0 as indicating no agreement and 0.01-0.20 as none to slight, 0.21-0.40 as fair, 0.41-0.60 as moderate, 0.61-0.80 as substantial, and 0.81-1.00 as almost perfect agreement. The inter-annotator agreement scores are presented in Table 5, the statistic underscores a substantial level of agreement among our human annotators, reaffirming the consistency and reliability of the annotations.

Table 5: Kappa agreement scores for annotation pairs

| Annotation Pair | Score |
|---|---|
| Cohen's Kappa (A1-A2) | 0.472 |
| Cohen's Kappa (A1-A3) | 0.698 |
| Cohen's Kappa (A2-A3) | 0.703 |
| Fleiss's Kappa (A1-A2-A3) | 0.614 |

Due to the page limitation, the experiments for "Effects of Different Generators" and "Case Study and Model Fine-tuning" are presented in Appendix.E and Appendix.F, respectively.

## 5 CONCLUSION

With the advanced natural language understanding capabilities of LLMs, existing textual attack methods often fail to affect them effectively. In this paper, we propose a novel textual attack method for generating adversarial examples that strike a balance between semantic preservation and sentiment manipulation. Our method employs carefully designed perturbation instructions to produce linguistically coherent and human-like adversarial texts. Additionally, a dual-layer hybrid filtering mechanism is introduced to ensure both semantic consistency and linguistic fluency. Extensive experiments conducted across multiple datasets and model architectures demonstrate that our method outperforms the SOTA baselines in terms of attack success rate while maintaining high text quality.

In the future, we will extend our work in two directions. Firstly, we plan to extend it to multilingual and code-mixed settings, explore controllable perturbation strength for flexible attack intensity. Secondly, we will investigate integrating sociolinguistic methods to enhance naturalness and develop defense strategies against implicit adversarial attacks.

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

## A    COMPLETE RELATED WORKS

Here, the completed related works w.r.t. textual adversarial attacks and evaluation paradigm of attacks are discussed.

### A.1    TEXTUAL ADVERSARIAL ATTACKS

According to the perturbation units imposed on the text, traditional methods can roughly be categorized into three types: character-level attacks, word-level attacks, and sentence-level attacks.

- Character-level attacks operate by introducing perturbations such as insertions, deletions, or substitutions at the character level, often exploiting vulnerabilities in tokenization or the sensitivity of models to spelling variations. For example, VertAttack (Rusert, 2024) exploits classifiers inability to read vertically written text by rewriting informative words in a vertical format, while Charmer (Rocamora et al., 2024) designs a method that leverages model gradient information and a position subset selection strategy to efficiently generate adversarial examples with minimal perturbations. Although these approaches are lightweight and efficient, they frequently yield unnatural or noisy text that can be easily detected by humans or spell checkers.

- Word-level attacks manipulate the input at the lexical level, typically by replacing or substituting words with semantically similar alternatives while attempting to preserve grammaticality and meaning. A representative example is HQA-Attack (Liu et al., 2023c), which proposes a black-box hard-label attack that iteratively substitutes original words back and optimizes with synonyms to achieve high semantic similarity and low perturbation rate under limited queries. Similarly, HyGloadAttack (Liu et al., 2024b) introduces a hybrid optimization framework with global substitution and a gradient-guided quick search, effectively mitigating local optima and reducing query overhead in hard-label black-box textual attacks. Compared with character-level perturbations, these methods usually produce more fluent adversarial samples, but they rely heavily on lexical resources and embedding quality, and may still introduce subtle semantic drift.

- Sentence-level attacks apply more substantial modifications, often involving paraphrasing or regenerating entire sentences to achieve adversarial effects. For instance, PI generates adversarial texts by inserting semantically meaningless parentheses and leverages a beam search strategy to maximize attack success while preserving semantics (Li et al., 2025). In contrast, recent studies exploit class probability feedback, with S2B2-Attack modeling adversarial distributions via variational autoencoders and applying score-based search to surpass blind and decision-based sentence-level attacks (Moraffah & Liu, 2024). These approaches can generate fluent and human-like adversarial samples, but they generally require more computation and risk introducing larger shifts in semantics compared to character- or word-level perturbations.

However, these methods typically require access to model internals or confidence scores, limiting their applicability to white-box or soft-label black-box scenarios. Beyond these, researchers have also explored semantic-level attacks, which aim to preserve fluency while subtly altering meaning. For example, paraphrasing techniques (Xu et al., 2024) aim to restate sentences with equivalent meaning but different wording, enhancing transferability. Cross-lingual transformations (Bhandari & Chen, 2023) leverage multilingual models to create adversarial examples by translating text between languages.

Another line of work emphasizes task-specific adversarial scenarios, such as evading toxic content detection, misinformation classification, or sentiment analysis systems (Bespalov et al., 2023; Przybyła et al., 2024). These studies highlight that adversarial text is not merely a technical challenge but also a significant threat to the secure deployment of NLP applications, particularly in domains where the stakes are high, such as content moderation and public opinion analysis. As large language models (LLMs) become more prevalent, they introduce new opportunities and risks, leading to the development of attacks targeting safety alignment and jailbreaking (Yi et al., 2024; Zhou et al., 2024). These emerging challenges underscore the need for more sophisticated attack strategies that go beyond simple perturbations, calling for techniques that can navigate the complexity of modern NLP systems.

To enhance the efficacy of attacks, a variety of optimization strategies have been proposed, including beam search (Zang et al., 2020), genetic algorithms (Liu et al., 2024b), reinforcement learning, and LLM-as-attacker paradigms. A promising new direction involves confidence elicitation techniques (Formento et al., 2025), which attempt to guide attacks by extracting calibrated confidence estimates from black-box models. However, most existing attacks still struggle to balance attack success, semantic coherence, and stealthiness.

Meanwhile, there is a growing body of research on defensive strategies, such as adversarial training, robust optimization, and consistency regularization (Zhao et al., 2024; Yang et al., 2024). While defenses reduce attack success rates to some extent, they often come at the cost of reduced model generalization. This tension further underscores the need for new attack paradigms that not only break existing defenses but also produce adversarial samples with higher semantic plausibility.

Building on this foundation, we propose a novel textual adversarial attack method that leverages linguistic patterns such as euphemism, sarcasm, and implicit sentiment expression. Our EuphemAttack operates in a black-box setting while maintaining both effectiveness and naturalness through prompt-based control and semantic filtering, addressing key limitations in existing attack strategies.

## A.2 Evaluation Paradigm of Attacks

Early approaches for natural language generation relied on surface-level lexical overlap metrics like BLEU (Papineni et al., 2002), ROUGE (Lin, 2004), and METEOR (Banerjee & Lavie, 2005). While effective for structured tasks such as summarization, these metrics fail to capture nuanced meaning shifts introduced by adversarial attacks. Other simple measures, such as edit distance or cosine similarity, provide complementary views but still lack semantic sensitivity.

With the development of pre-trained models like BERT and BART, model-based evaluation paradigms emerged, enabling assessment of relevance, consistency, and factuality through contextualized embeddings (Zhang et al., 2019; Yuan et al., 2021). These approaches significantly improved semantic alignment but still exhibited limitations in capturing subtle pragmatic shifts, which are often crucial in adversarial scenarios. More importantly, many existing evaluations only focus on attack success rate or perturbation size, while overlooking adversarial properties such as stealthiness, cross-model transferability, and emotional subtlety.

Recently, LLMs have been used as evaluators to assess generated text with high agreement to human judgments (Gu et al., 2024). Frameworks such as G-EVAL introduce an LLM-as-a-judge paradigm that combines automatic chain-of-thought, form-based scoring, and probability-weighted aggregation to improve alignment with human judgments across summarization and dialogue tasks (Liu et al., 2023d). Similarly, JudgeLM fine-tunes open-source language models with swap augmentation, reference support, and reference drop to mitigate position, knowledge, and format biases, while also adopting a score-first paradigm to enhance evaluation efficiency (Zhu et al.). Beyond these, EvalPlanner proposes splitting evaluation reasoning into planning, execution, and judgment, leveraging unconstrained, self-training data to improve transparency and performance across diverse tasks, thus reducing reliance on manual component design (Saha et al.).

Despite these advances, challenges remain. Lu et al. present a systematic benchmark of closed- and open-source LLMs on offensive language detection with annotation-disagreement samples, revealing performance degradation and overconfidence in ambiguous cases (Lu et al., 2025). Their findings suggest that incorporating disagreement data in few-shot and fine-tuning settings can improve generalization and alignment with human judgments. More broadly, LLM evaluators continue to face persistent issues such as subjectivity, sensitivity to prompt design, and potential bias, especially in adversarial contexts. The inherent complexity of adversarial attacks and the limitations of current evaluation methods often necessitate additional verification, which can be resource-intensive. This underscores the need for more refined frameworks that can effectively capture subtle semantic shifts in adversarial examples without incurring excessive validation costs.

Specifically, we employed large language models to evaluate not only the attack success rate but also semantic preservation, emotional subtlety, and linguistic naturalness of adversarial outputs. Multiple advanced models were utilized for semantic evaluation, thereby bridging the gap between quantitative metrics and qualitative assessments and providing a more comprehensive benchmark for textual adversarial attacks.

## B    THE CONSTRUCTED PROMPTS IN OUR EUPHEMATTACK

Here, the detailed prompts in Section 3.3.1 PROMPT CONSTRUCTION are presented.

**(1) The Prompts of Attack Objective:**

---
**# Attack Objective**

Your task is to generate a new sentence that must meet the following conditions:
1. Keep the semantic meaning of the new sentence unchanged in the sentence.
2. The new sentence should be classified as $\{target\_label\}$.
3. The new sentence should conform to the expression habits of natural human language.

---

**(2) The Prompts of Entity Retention:**

---
**# Entity Retention**

You must strictly retain all original entities (including names of people, places, organizations, brands, product names, and any other concrete nouns) exactly as they appear in the $\{X\}$.
1. You must not omit, substitute, modify, generalize, or add new entities.
2. The semantic relationships involving these entities must remain unchanged.

---

**(3) The Prompts of Perturbation Guidance:**

---
**# Perturbation Guidance**

If $X$ contains expressions that convey strong positive or negative emotions, you must rewrite the sentence to weaken or hide these emotional expressions. The transformation should employ:

- If $\{ground\_turth\}$ is positive, rewrite the sentence in an understated, calm, or indifferent tone.
- If $\{ground\_turth\}$ is negative, rewrite the sentence in an euphemistic, implicit, or subtly sarcastic tone.

Commonly used linguistic techniques such as references to historical or cultural contexts, idioms, proverbs, metaphors, or colloquial expressions where appropriate and consistent with the sentence context.

---

## C    THE STATISTICS OF DATASETS AND EXPERIMENTAL SETTING

### C.1    THE STATISTICS OF ALL THE DATASETS

The statistics of all the datasets are presented in Table 6.

Table 6: The statistics of all datasets. "Pos.Num" and "Neg.Num" represent the number of positive and negative instances, respectively.

| Dataset | DataSize | Pos.Num | Neg.Num |
|---------|----------|---------|---------|
| IMDB | 4000 | 2000 | 2000 |
| Amazon | 4000 | 2000 | 2000 |
| Yelp | 4000 | 2000 | 2000 |
| DianPing | 5015 | 2589 | 2426 |
| CTrip | 9426 | 6213 | 2313 |
| JD | 4673 | 2324 | 2343 |

### C.2    EXPERIMENTAL SETTING

Our method employs a few-shot learning approach, utilizing 5 carefully selected positive-negative sample pairs to guide the model's sentence generation process. To ensure the robustness and quality of our adversarial examples, we generated three distinct sets of adversarial samples using the same

generation strategy. In this design, the guidance examples help the model retain the core semantics of the input while naturally embedding subtle stylistic cues aligned with the target label. To reduce potential variance in single evaluations and obtain more reliable results, we conducted three independent generations for each evaluation and reported the average performance.

Unlike many existing methods that rely on heuristic or beam search with extensive queries, our EuphemAttack fully leverages the model's generative ability by simply generating a few candidates and selecting the optimal adversarial sample, the query cost remains minimal while still achieving strong attack performance. For the baselines we employed the default hyperparameters reported in the original papers. Most baseline methods considered are black box hard label attacks that perform word replacements based on model decisions or on model scores via extensive model queries. PromptAttack is an exception and operates in a few-shot setting similar to our method. To assess transferability, for each baseline we generated adversarial examples under its default configuration that yielded the best performance, then used those examples to attack the target models evaluated in this study and compared those results to the performance of our method.

All experiments were conducted on a high-performance server featuring an NVIDIA GeForce RTX 4090 Founders Edition GPU, coupled with a 13th Gen Intel(R) Core(TM) i9-13900K CPU, operating at a frequency of 3.00 GHz, and equipped with 125 GB of RAM. The experiments utilized Python version 3.9.16, PyTorch framework, and CUDA version 12.1 to leverage GPU acceleration.

## D    EVALUATION WITH HUMANS

The undergraduate students are experienced users of major social media platforms, with over five years of account registration history and more than 20 hours of weekly activity. This extensive usage enables them to develop a deep understanding of comments commonly found on these platforms. As shown in the Table 7, we list the information of the annotators.

Table 7: Demographic and Activity Profile of Annotators

| Total Annotators | Gender (M/F) | Age Range | Avg. Registration Time | Avg. Weekly Active Hours |
|---|---|---|---|---|
| 3 | 2 / 1 | 22 – 25 yrs | >5 years | >20 hours |

During the evaluation process, we initially annotated 1,000 data samples jointly by all authors and inserted them randomly into the original dataset without the annotators' knowledge. After completing the annotation, we compared these 1,000 data samples with the original labels. If there was a significant deviation, indicating unreliable annotation, we replaced it with annotations from a different annotator.

## E    INFLUENCE OF DIFFERENT GENERATORS

To evaluate the robustness of our EuphemAttack with respect to the selection of generator, we further conducted experiments using two additional LLMs, Gemini and Qwen3. These models differ in architecture and training data scale from the primary generator used in our main experiments, thus providing a complementary perspective on the stability of the proposed approach. Specifically, we replaced the generator in our attack pipeline with Gemini-2.5-flash and Qwen3-235b-a22b while keeping the victim models and evaluation settings unchanged. The goal is to assess whether variations in the generator affect the attack success rate (ASR) and the drop in accuracy.

As shown in Table 8, although different generators lead to slight fluctuations in ASR and accuracy reduction, the overall trend remains consistent. Gemini tends to generate more diverse paraphrases, resulting in slightly higher ASR, whereas Qwen achieves more fluent but less aggressive perturbations, leading to slightly lower ASR. These results demonstrate that the effectiveness of our EuphemAttack is not tied to a single generator, and the attack maintains strong transferability across different LLMs.

Overall, these findings suggest that our EuphemAttack is generator-agnostic to a large extent. While different LLMs bring stylistic and lexical variations, the euphemism-based rephrasing strategy reliably introduces semantic-preserving perturbations that can fool victim models. This indicates that

Table 8: Influence of different generators on all the datasets. For English datasets, adversarial examples were generated using Gemini-2.5-flash, while for Chinese datasets, Qwen3-235b-a22b was employed. **Ori.%** represents the original accuracy, **Acc.%** indicates the accuracy after adversarial attacks, and **ASR%** denotes the attack success rate, with the values in parentheses indicating the change relative to the main experiments (↑ for increase, ↓ for decrease).

| Generator | Victim | Datasets | Ori.% | Acc.% | ASR% |
|---|---|---|---|---|---|
| Gemini | GPT-4o | IMDB | 95.25 | 76.11 | 20.09(8.87↑) |
| | | Amazon | 95.27 | 78.85 | 17.23(7.3↑) |
| | | Yelp | 97.67 | 81.26 | 16.80(10.76↑) |
| | Llama | IMDB | 94.04 | 70.83 | 24.68(3.27↑) |
| | | Amazon | 94.65 | 72.84 | 23.04(3.03↓) |
| | | Yelp | 96.29 | 75.34 | 21.76(5.09↓) |
| Qwen3 | DeepSeek | DianPing | 78.21 | 61.34 | 21.57(0.77↓) |
| | | CTrip | 82.29 | 69.87 | 15.09(0.06↓) |
| | | JD | 93.80 | 72.39 | 22.82(0.11↓) |
| | Qwen | DianPing | 76.09 | 59.94 | 21.23(1.94↓) |
| | | CTrip | 83.87 | 67.52 | 19.49(2.93↑) |
| | | JD | 93.84 | 70.92 | 24.42(7.2↓) |
| | GLM | DianPing | 75.83 | 58.34 | 23.07(1.38↑) |
| | | CTrip | 81.68 | 65.64 | 19.64(8.65↑) |
| | | JD | 92.66 | 65.47 | 29.34(0.76↓) |

future advances in LLMs could further enhance the quality of generated adversarial examples without undermining the core mechanism of our method.

# F CASE STUDY AND MODEL FINE-TUNING

## F.1 CASE STUDY

In Table 9, we present a comparative analysis between our EuphemAttack and three variants of PromptAttack across different domains. These cases further reveal significant limitations in these baseline approaches. While these methods attempt to generate adversarial examples, they often struggle to maintain semantic consistency with the original text, resulting in either semantically altered content or invalid adversarial samples that fail to meet semantic perturbation constraints. In contrast, our EuphemAttack demonstrates superior performance in terms of coherence, conciseness, and readability. More importantly, it successfully achieves a delicate balance between semantic preservation and effective semantic perturbation. This balance is crucial for generating high-quality adversarial examples that can effectively test model robustness while maintaining the original meaning and natural language characteristics. The case study further illustrates how our EuphemAttack can subtly alter sentiment while preserving the core message and maintaining grammatical correctness, making it particularly suitable for real-world applications where semantic integrity is paramount.

## F.2 MODEL FINE-TUNING

As shown in Fig. 3, we conducted fine-tuning experiments on both Bert and Llama across various English and Chinese datasets, where the adversarially attacked datasets were split into training, validation, and test sets with a ratio of 8:1:1. Bert and Llama are treated as the representative models of PLMs and open-source LLMs. For Llama, we employed LoRA-based fine-tuning to enhance efficiency. The results demonstrate that our EuphemAttack, while relatively simple, is highly effective in significantly degrading model performance. Notably, when we applied the same adversarial attacks to the fine-tuned models, all accuracy metrics showed significant improvements, with Llama achieving an impressive 92.08% accuracy even in Chinese language tasks where it traditionally underperforms. The fine-tuning process proves to be a robust defense mechanism, effectively mitigating the influence of our adversarial attacks. This highlights the dual nature of our EuphemAttack: its capability to expose vulnerabilities in language models and the importance of incorporating ad-

Table 9: Case study comparing our method with PromptAttack (1: adding meaningless characters, 2: paraphrasing, 3: changing syntactic structure).

| Method | Sentence | Prediction |
|---|---|---|
| Original | Its cozy vibe and rich flavors won my heart thoroughly. | **Positive→Negative** |
| PromptAttack-1 | Its cozy vibe and rich flavors won my heart thoroughly. @r4xmn2 | |
| PromptAttack-2 | I completely fell in love with its warm atmosphere and delicious taste. | |
| PromptAttack-3 | Thoroughly won over by its rich flavors and cozy vibe, my heart couldn?t resist. | |
| Ours | Its cozy vibe and rich flavors ... well, let's just say they didn't go unnoticed. | |
| Original | A messy plot with flat acting left me yawning nonstop. | **Negative→Positive** |
| PromptAttack-1 | A messy plot with flat acting left me yawning nonstop. @qezro1 | |
| PromptAttack-2 | The chaotic plot and dull acting kept me fighting off yawns throughout. | |
| PromptAttack-3 | Yawning nonstop was the result of a flat performance combined with a messy plot. | |
| Ours | The plot's loose threads with understated acting settled me in quiet ease. | |

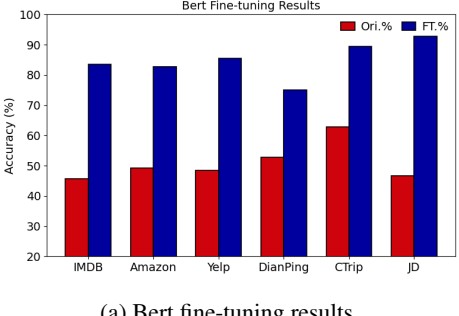

(a) Bert fine-tuning results

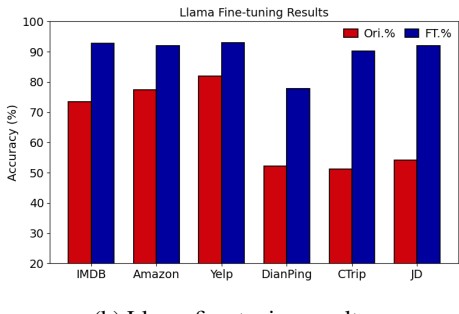

(b) Llama fine-tuning results

Figure 3: Fine-tuning results on Bert and Llama across all datasets, showing original accuracy (Ori.%) and accuracy after fine-tuning (FT.%). Upward arrows indicate performance improvement after fine-tuning.

versarial samples into training data as a countermeasure to enhance model resilience against such attacks.

