# OpenReview forum: "Can LLMs be Fooled: A Textual Adversarial Attack method via Euphemism Rephrase to Large Language Models"
_ICLR.cc/2026/Conference — ICLR 2026 Conference Withdrawn Submission_

### Official Review · Reviewer_fi9D · 2025-10-20

**Soundness:** 2
**Presentation:** 2
**Contribution:** 2
**Rating:** 4
**Confidence:** 3

**Summary:**

The authors find that the previous textual adversarial attacks, including character-level, word-level, and simple sentence-level paraphrasing, are largely ineffective against modern LLMs. Therefore, this paper proposes a new textual adversarial attack that aims to generate adversarial examples with euphemistic and ironic style. This method uses a few-shot prompting strategy to guide an LLM for generating adversarial examples and uses a hybrid filter to select good demonstration examples. The authors conduct both automated experiments on different sentiment analysis datasets and human evaluation to prove that their method can not only achieve superior attack success rate but also effectively preserve the original semantics.

**Strengths:**

1. The adversarial robustness of the current LLMs is an important topic to study.

2. The paper is well-organized and easy to follow.

3. The authors reveal that LLMs are vulnerable to the adversarial examples with the euphemistic style and the proposed attack method shows superior attack performance over the baselines on the sentiment analysis task.

**Weaknesses:**

1. The adversarial attack proposed in the paper mainly focuses on the sentiment analysis task. It is unknown whether it is applicable to other NLP tasks like QA and coding.  It seems the attack effectiveness requires the careful design of prompt instruction for the generator, which is not a general, automated method.

2. More experiments should be done to consolidate the designs:

a. No ablation study has been done in the paper. The authors should show the effect of the hybrid filter/number of shots/perturbation guidance on the attack success rate.

b. Although the author conducts human evaluation on the semantics of adversarial examples, the accuracy of human annotators in classifying the adversarial examples is not reported (assuming the target label is the ground truth).

c. Although the authors try Gemini and Qwen3-235b-a22b as the generator in Appendix E, it is still unknow how well models with smaller sizes can perform.

3. The paper presentation can be improved.

a. The concrete generator used in the main paper is not mentioned.

b. Table 1 and Table 2 should be fit within the line.

**Questions:**

1. What is the difference between the prompt attack in Table 1 and those in Table 3?

2. Would the euphemistic style be helpful for adversarial attacks in other NLP tasks?

3. Apart from the euphemistic style, is there any other linguistic style which can function similarly for adversarial attack purpose?

---

### Official Review · Reviewer_Bwyz · 2025-10-28

**Soundness:** 3
**Presentation:** 3
**Contribution:** 2
**Rating:** 2
**Confidence:** 3

**Summary:**

This paper introduces EuphemAttack, a novel textual adversarial attack method designed to overcome the robustness of modern Large Language Models (LLMs) where traditional attacks fail. The method leverages LLMs' weakness in understanding nuanced language, guiding a generator model via perturbation instructions to rephrase text into implicit, euphemistic, or ironic expressions. This approach aims to alter the model's sentiment classification while preserving the original semantic meaning and entity consistency. A dual-layer hybrid filter, integrating BERTScore for similarity and perplexity for naturalness, is employed to ensure high-quality output. Experiments on SOTA LLMs like GPT-4o and Llama demonstrate EuphemAttack's significantly higher Attack Success Rate (ASR). Furthermore, comprehensive evaluations by both LLMs and human annotators validate its superiority in maintaining text quality, coherence, and naturalness over existing methods.

**Strengths:**

1 The paper's contribution is of high originality and significance. It moves beyond traditional lexical perturbations to pioneeringly identify a novel, deeper vulnerability: LLMs' deficiency in understanding pragmatics, specifically euphemism and sarcasm .
2 The methodological quality is high. The EuphemAttack framework systematically operationalizes this idea via a creative three-component constrained prompt (notably "Entity Retention" and "Perturbation Guidance") and a dual-layer hybrid filter , successfully balancing semantic fidelity with stylistic manipulation.
3 The evaluation is exceptionally rigorous and comprehensive. It validates effectiveness not only with Attack Success Rate (ASR) metrics but also with meticulous textual quality assessments using both "LLM-as-a-judge" and (reliable) human evaluation .
4 The paper's credibility is substantially reinforced by calculating Kappa inter-annotator agreement (Table 5) . This (often-omitted) step is critical for validating subjective assessments and significantly enhances the clarity and trustworthiness of the findings.

**Weaknesses:**

1 The paper's experimental validation is limited to a single type of task and dataset (primarily product and reviews ). This leaves the method's generalizability unproven, and it is unclear if EuphemAttack would be effective against more diverse NLP tasks.
2 The paper suffers from reproducibility concerns. The authors omit several critical methodological details, such as the specific construction or selection criteria for the few-shot examples used for guidance , and the key hyperparameter settings for the quality filter.
3 The paper's claim of being "generator-agnostic to a large extent" is an overstatement. The experimental data in the appendix (Table 8 ) clearly shows that the Attack Success Rate (ASR) is highly dependent on the generator model used, leading to significant variations in attack performance.

**Questions:**

1 The paper claims that euphemistic rephrasing can deceive LLMs where character-, word-, or sentence-level attacks fail. However, the introduction provides little theoretical or empirical justification for why euphemism, among many linguistic devices (e.g., irony, metaphor, understatement), should be particularly effective. Could the authors clarify the linguistic or cognitive rationale that links euphemistic or implicit expression to model vulnerability? Without such grounding, the motivation appears largely post-hoc.
2 The experiments demonstrate increased ASR, but they do not isolate the specific effect of euphemism. Have the authors conducted any controlled studies comparing euphemistic rephrasing with other stylistic or paraphrasing strategies to confirm that the improvement originates from euphemistic tone rather than general paraphrase fluency or prompt variance? Moreover, the proposed hybrid filter (BERTScore + PPL) primarily enforces fluency, not adversariality—how do the authors verify that the resulting sentences truly preserve semantics while misleading the model?
3 The paper does not discuss prior rephrase-based adversarial frameworks such as LLM-Fuzzer (USENIX 2023), which already explored reformulation-based jailbreaks. How does EuphemAttack differ methodologically and conceptually from this line of work? Additionally, given that the method depends on LLMs to generate and filter attacks, could the authors elaborate on efficiency, scalability, and reproducibility issues, as well as reconcile this dependence with the claim of being “generator-agnostic”?

---

### Official Review · Reviewer_Embg · 2025-11-01

**Soundness:** 2
**Presentation:** 2
**Contribution:** 2
**Rating:** 2
**Confidence:** 5

**Summary:**

This paper addresses the declining effectiveness of conventional textual adversarial attacks (e.g., character/word/sentence-level perturbations) against modern LLMs such as GPT-4. The authors propose EuphemAttack, a euphemism-based adversarial attack method that subtly hides sentiment cues while keeping semantic content and entities intact. The approach uses prompt-based perturbation guidance and a hybrid filtering mechanism combining BERTScore and perplexity to ensure semantic preservation and linguistic naturalness. Experiments on multiple English and Chinese sentiment classification datasets show that EuphemAttack significantly improves the attack success rate over existing methods while producing more fluent, coherent adversarial examples. Automatic and human evaluations both confirm stronger text quality compared to baselines.

**Strengths:**

1. The proposed euphemism-based attack strategy produces adversarial examples with significantly better coherence, fluency, and semantic consistency compared to multiple attacks. These high-quality examples are also successfully deceive multiple LLMs across English and Chinese settings, demonstrating strong general effectiveness.

2. The paper conducts thorough human assessment on multiple dimensions (coherence, fluency, grammaticality, naturalness), which enhances the credibility of results and strengthens claims about semantic preservation.

**Weaknesses:**

1. The paper title currently contains a stray phrase (“CONFERENCE SUBMISSIONS”), which signals a lack of careful proofreading. While minor, such presentation problems may reduce the perceived polish and professionalism of the work, particularly for a top-tier venue like ICLR.

2. The paper focus solely on sentiment classification — a task that modern instruction-following LLMs generally handle robustly and that is less central in current LLM security research.

3. Victim models are insufficient and lack clarity: Although multiple LLMs are attacked, the work does not include the strongest reasoning LLMs (e.g., OpenAI o3, Anthropic Claude 4.5).The specific DeepSeek model version is not clearly identified, which directly affects reproducibility and evaluation reliability.

4. Missing analysis on generator–victim capability gap: The attack heavily relies on LLMs as generators, yet the paper does not examine a key real-world scenario: Can adversarial samples crafted by a weaker model effectively transfer to much stronger victim models? This is critical because: If transferability significantly drops against stronger targets, the threat model becomes limited. A systematic study on cross-model transfer, especially across different capability tiers, is necessary to validate the method’s robustness and applicability.

**Questions:**

1. Have the authors tested whether existing defense techniques (e.g., consistency-based defenses, prompt engineering) can mitigate the proposed attack? This would indicate true robustness and practical relevance.

2. How many queries or generation attempts are required per successful attack? An explicit cost evaluation is needed to assess feasibility in real deployment.

3. Few-shot defense feasibility: If the victim model is provided with a few adversarial examples demonstrating euphemistic patterns, can it quickly learn to resist the attack? This relates to safety alignment stability and adaptation.

---

### Official Review · Reviewer_xDNP · 2025-11-15

**Soundness:** 4
**Presentation:** 4
**Contribution:** 2
**Rating:** 4
**Confidence:** 2

**Summary:**

The paper studies textual adversarial attack strategies. Existing character-level, word-level, and sentence-level attacks are no longer effective against large language models (LLMs).
To address this, the paper introduces an adversarial attack method named EuphemAttack, which generates euphemistic rephrasings that can deceive LLMs without altering the original meaning and while remaining understandable to humans.

The proposed method first generates linguistically coherent and human-like adversarial examples, then applies a dual-layer hybrid filter to ensure both semantic similarity and linguistic naturalness.
The goal is to rephrase the original text into implicit, euphemistic, or ironic expressions that are common in everyday language—maintaining semantic fidelity and entity consistency while subtly altering sentiment cues to mislead LLMs.

Experiments are conducted on GPT-4 and DeepSeek, among other LLMs.
Evaluations on coherence, fluency, grammar, and naturalness show that the proposed EuphemAttack better preserves text quality compared with other attack methods.

**Strengths:**

The paper is well written.

The proposed method is clearly described and appears easy to implement.

The paper also provides comprehensive experimental results.

**Weaknesses:**

The criteria for choosing thresholds for semantic similarity and acceptable perplexity are not clearly explained.

Some related work is missing, such as Phrase-level Textual Adversarial Attack with Label Preservation (NAACL 2022).

It is also unclear which model is used to generate adversarial examples, and how the results might differ when using different models.
Does the paper evaluate multiple models?

In addition, the experimental scope is narrow — only polarity classification tasks are presented.

**Questions:**

The effectiveness of the prompt-based method may depend on the specific model used.
Does the paper compare the performance across different models?

---

### Note · Authors · 2025-11-21

I have read and agree with the venue's withdrawal policy on behalf of myself and my co-authors.